# Psychophysiological Responses in Soldiers during Close Combat: Implications for Occupational Health and Fitness in Tactical Populations

**DOI:** 10.3390/healthcare12010082

**Published:** 2023-12-29

**Authors:** Maria Stergiou, José Juan Robles-Pérez, Jorge Rey-Mota, José Francisco Tornero-Aguilera, Vicente Javier Clemente-Suárez

**Affiliations:** 1Faculty of Sports Sciences, Universidad Europea de Madrid, 28670 Madrid, Spain; vicentejavier.clemente@universidadeuropea.es; 2Center for Applied Combat Studies (CESCA), 45007 Toledo, Spain; jrobper@et.mde.es (J.J.R.-P.); jreymota@gmail.com (J.R.-M.); 3USAC ‘San Cristobal-Villaverde’—Ejército de Tierra, Av. de Andalucía, Km. 10, Villaverde, 28021 Madrid, Spain; 4Grupo de Investigación en Cultura, Educación y Sociedad, Universidad de la Costa, Barranquilla 080002, Colombia

**Keywords:** melee combat, psychophysiological stress, rate of perceived exertion, cortical arousal, military training

## Abstract

This study explores the psychophysiological responses and subjective exertion experiences of soldiers in simulated hand-to-hand combat, aligning these findings with established physiological benchmarks. Active military personnel were monitored for heart rate, blood lactate levels, subjective exertion, cortical arousal, and muscle strength during combat scenarios. The results showed significant increases in heart rate and blood lactate, indicating intense cardiovascular demands and a reliance on anaerobic energy systems. Contrary to these physiological changes, soldiers reported lower levels of exertion, suggesting a possible underestimation of physical effort or individual differences in perception and mental resilience to stress. Notably, a decrease in cortical arousal post-combat was observed, potentially signaling cognitive function deficits in decision-making and information processing in high-stress environments. This decline was more pronounced than typically seen in other high-stress situations, highlighting the unique cognitive demands of hand-to-hand combat. Additionally, an increase in muscle strength was noted, underscoring the physiological adaptations arising from intensive combat training. These findings provide valuable insights into the psychophysiological effects of hand-to-hand combat, emphasizing the complex interplay between physical exertion, cognitive function, and stress response in military contexts. The study underscores the need for comprehensive training strategies that address both physical and psychological aspects to enhance combat readiness and decision-making under stress.

## 1. Introduction

In contemporary conflict environments, military personnel are subjected to a multifarious array of combat modalities. The traditional paradigm of symmetrical warfare, characterized by well-defined front lines and identifiable combatants, coexists with asymmetrical warfare. The latter often unfolds in densely populated urban settings, involves non-combatants, and is marked by rapidly evolving tactical scenarios. This complexity necessitates that soldiers possess proficiency not only in conventional battlefield tactics but also in navigating unpredictable close-quarters and melee combat situations. Particularly, the prevalence of insurgent assaults on military contingents, frequently occurring at checkpoints and bases, requires soldiers to adeptly transition between diverse combat styles while adhering to stringent rules of engagement.

Consequently, scholarly research into the physiological and psychological ramifications of combat stress has been extensive, yielding a nuanced comprehension of its impact on soldiers in combat scenarios. A synthesis of pertinent literature [1,2,3,4] reveals that combat stress exerts a significant influence on the sympathetic nervous system, manifesting in elevated metabolic rates, increased muscular tension, and heightened cardiovascular activity, coupled with a decrease in cortical arousal. These physiological adaptations, while evolutionarily advantageous for immediate survival responses, have been documented to adversely impact critical higher-order cognitive functions essential in combat, such as decision-making and memory retention [5]. Furthermore, the complexity of soldiers’ perceived exertion under stress is underscored, often resulting in a dichotomy of either overexertion or underutilization of their physical and cognitive capacities in high-stakes situations [2]. This facet of combat stress accentuates the imperative for more sophisticated training methodologies that encompass both physical endurance and cognitive resilience.

The inherent unpredictability of melee combat, characterized by proximate, hand-to-hand engagements, significantly intensifies stress responses. These reactions, far more acute and distinct compared to more predictable or controlled combat environments, are critical in understanding the psychophysiological dynamics of such scenarios. The complexities of acute stress reactions (ASRs), encompassing a broad spectrum of psychological and physiological symptoms during high-stress events, have notable implications for operational functionality, particularly in professions such as the military, policing, firefighting, and emergency medicine.

This understanding is crucial for developing effective training programs and preparing soldiers psychologically for the demands of modern warfare. Moreover, the implications of chronic exposure to high-stress environments extend beyond immediate tactical considerations, impacting long-term health outcomes. Persistent activation of the sympathetic nervous system, a response to continuous stress, has been linked to an increased risk of cardiovascular pathologies [6,7]. This link, detailed in the work of Schneiderman, Ironson, and Siegel [8], emphasizes the vital connection between prolonged stress exposure and cardiovascular health. It reinforces the need for comprehensive combat stress management strategies, not only for immediate operational efficiency but also for the sustained health and welfare of military personnel. These strategies are integral to maintaining the long-term readiness and resilience of those serving in high-stress occupations.

In this context, our study endeavors to conduct an exhaustive investigation into the psychophysiological alterations experienced by soldiers in simulated melee combat scenarios. By examining a spectrum of variables, including but not limited to cortical arousal, autonomic modulation, blood lactate concentration, muscle strength, heart rate, and perceived exertion, we hypothesize that simulated melee combat scenarios will precipitate a marked amplification of physiological responses and perceived exertion, concomitant with a significant diminution in cortical arousal. This research is positioned to substantially augment the extant corpus of knowledge, providing deeper insights into the complex psychophysiological dynamics of combat. Our study is uniquely situated to elucidate these dynamics in the unpredictable and high-intensity milieu of melee combat. The anticipated findings are poised to have considerable implications, contributing to the development of more efficacious and holistic training and support frameworks for soldiers in high-stress combat roles, thereby enhancing their immediate operational effectiveness and long-term psychological resilience.

## 2. Materials and Methods

### 2.1. Participants

For this study, an opportunity sample of 50 soldiers were chosen for participation. These individuals are routinely subjected to highly stressful situations, yet due to the specialized nature of their profession, the pool of potential participants is inherently limited, resulting in a relatively small sample size. The participants were members of the Army and State Security Corps, with an average age of 35.9 years, an average height of 177.2 cm, a weight of approximately 75.4 kg, a muscular mass of 64.1 kg, and a fat mass of around 8.3 kg. These soldiers, characterized by their extensive field experience, typically served in roles that demanded high adaptability, strategic acumen, and resilience. Their combat histories, ranging between 7 and 18 years, encapsulate not only duration but also the intensity and diversity of military engagements. In international conflict zones such as Lebanon, Afghanistan, Bosnia, Kosovo, and Iraq, these individuals often encountered varied combat scenarios—from guerrilla warfare in rugged terrains to urban warfare amidst densely populated cities. Their experience includes but is not limited to counter-insurgency operations, peacekeeping missions under the UN or NATO mandates, and direct combat against organized military forces. Many have been involved in joint operations with allied forces, requiring them to adapt to different command structures, engagement rules, and cultural sensitivities. They have often operated in environments marked by extreme weather conditions, hostile locals, and challenging logistical scenarios.

As inclusion criteria, individuals were required to have a solid background in active combat roles or be in support positions that directly impacted the outcomes of battles in international conflict zones. They needed to have at least 7 years of total military service, with a significant portion of that time spent in active conflict zones. Their experience should not have been limited to a single region; participation in combat across at least two different international conflict zones was mandatory, highlighting their adaptability to different warfare environments and situations. Additionally, these soldiers were expected to have experience in joint operations, indicating their proficiency in collaborating within multinational military frameworks and adapting to various command structures and tactics. Official recognition or commendations for their performance and conduct in these zones were also critical, as this reflects their professionalism and effectiveness in challenging situations.

As exclusion criteria, individuals primarily engaged in non-combat roles such as administrative, logistical, or other support functions without direct combat exposure were excluded. Soldiers whose service in conflict zones was insufficient, particularly those serving less than the required duration in such environments, were not considered. Any history of disciplinary actions or court-martials for misconduct, especially during service in conflict zones, disqualified a candidate. Similarly, those lacking experience in multinational or joint operations, crucial for understanding diverse military strategies and environments, were not eligible. Lastly, soldiers had to be physically and psychologically fit for duty in demanding conditions; those who were medically unfit or incapable of performing in strenuous or hostile settings were excluded from this group. These criteria ensured that the selected soldiers were not only experienced but also had demonstrated adaptability, resilience, and professional excellence throughout their military careers.

Furthermore, this investigation was conducted in strict alignment with the ethical principles outlined in the Declaration of Helsinki. All participants were fully briefed on the study’s nature and provided informed consent. The experimental protocol received formal approval from the Unit’s Head Quarter and the European University of Madrid ethics committee CIPI/18/093. We instructed the participants to execute the exercise with the same precision and commitment they would apply in an actual combat situation, ensuring the data’s relevance and applicability to real-world scenarios.

### 2.2. Design and Procedure

The study utilized a small group of soldiers, employing a cross-sectional pre-post design to limit the influence of individual variances on psychophysiological changes observed. Data collection was conducted prior to and immediately after the drill.

The primary variable under investigation was the timing of the assessment, categorized into two distinct phases: prior to and following a close-quarters combat drill. This drill was uniquely designed, replicating authentic scenarios akin to those encountered in actual conflict zones such as Eastern Europe. The exercise involved 50 soldiers, grouped into 10 groups of 5 soldiers, tasked with a defensive and protective operation. The simulation presented them with two different adversaries: one, an organized and armed combatant, and the other, an unarmed and disorganized potential threat, likely to attempt theft at the soldiers’ base. The teams were challenged to evaluate these threats and react as per engagement rules, choosing either to fire upon or subdue the adversaries without using their weapons.

Prior to the drill, an integrated combat readiness training protocol (ICRTP) of 10 min was applied. A holistic training program was specifically designed to prepare soldiers for the intense physical, cognitive, and psychological demands of modern combat. The protocol was a blend of high-intensity interval training (HIIT), cognitive load exercises, and stress inoculation training (SIT), strategically sequenced to optimize soldiers’ overall combat effectiveness.

The ICRTP commences with a dynamic sequence of HIIT exercises. These exercises involve short bursts of high-intensity activities such as sprinting, burpees, and jump squats, interspersed with brief recovery periods. The aim here is to rapidly escalate heart rate to near-maximum, thereby simulating the intense physical exertion required in combat. Then, as the heart rate peaks following HIIT, soldiers immediately transition into cognitively demanding tasks, ensuring no cooling-down period. This direct shift from physical exertion to mental exertion is critical to the protocol. These tasks include complex problem-solving exercises, quick-recall memory tests, and on-the-spot tactical decision-making drills. These tasks are carried out in a time-sensitive and high-pressure setting. Finally, on completion of the cognitive phase, soldiers are immediately immersed in a stress inoculation training section. This segment involves navigating through disorienting environments (e.g., smoke-filled rooms), reacting to sudden loud stimuli mimicking battlefield noises, and handling high-pressure situations such as aggressive interrogations or mock enemy engagements.

Then, when the ICRTP was finished, soldiers would rush towards a designated building where the adversary was presumed to be. On entering the building, the soldiers confronted an adversary wearing Redman protective gear, which shields the wearer from injury, imitating an opponent who is impervious to pain, reflecting the toughness of insurgents in real combat zones. The scenario unfolded in dimly lit conditions, compelling the soldiers to apprehend the enemy without resorting to firearms. This was in accordance with the rules of engagement, which dictated handling the confrontation through hand-to-hand combat techniques, as the opponent was unarmed.

### 2.3. Materials

To investigate the impact of the combat simulation after acute psychophysiological stress, several key dependent measures were recorded 1 h before and immediately after the exercise. The enhancements and details of the original measures are as follows:

Perceived exertion rating: We used Borg’s 6–20 scale [9], a well-established psychometric tool for evaluating participants’ subjective effort and exertion levels. This scale ranges from 6, indicating no exertion, to 20, representing maximal exertion. Its design approximates the heart rate divided by 10; thus, a rating of 12 should correspond to a heart rate of approximately 120 bpm. The Borg’s 6–20 scale has been validated across various studies, showing a strong correlation with physiological markers of exertion such as lactate concentration and oxygen uptake, with correlation coefficients typically ranging from 0.6 to 0.8. This indicates a high level of concurrent validity. Additionally, the scale demonstrates high test–retest reliability, ensuring consistent results across multiple assessments. Such psychometric properties make the Borg’s 6–20 scale a reliable and valid tool for measuring perceived exertion in our study, providing a crucial subjective complement to the objective physiological data collected.

Cortical arousal was assessed using the critical flicker fusion threshold (CFFT). An increase in CFFT indicates heightened cortical arousal and improved information processing capabilities, while a decrease suggests reduced processing efficiency and central nervous system fatigue [4,10,11]. For this assessment, subjects were positioned in front of a viewing chamber (Lafayette Instrument Flicker Fusion Control Unit Model 12021), designed to mitigate external factors affecting CFFT readings. Inside the chamber, subjects were exposed to two light-emitting diodes (58 cd/m^2^), one for each eye, positioned 2.75 cm apart (center to center) at a 15 cm distance and a viewing angle of 1.9°. The chamber’s interior was painted flat black to reduce reflections. The test involved increasing the flicker frequency from 20 to 100 Hz in 1 Hz/s increments until the participant perceived a steady light, indicating fusion. Participants underwent a fovea binocular fixation and then pressed a button to signal the perception of visual fusion. To ensure familiarity with the test procedure, participants engaged in several practice trials before the actual test. The test itself consisted of five trials, each separated by a 5 s interval. The time taken by a subject to detect the change in light from flicker to fusion was recorded for each trial, and the average of these times was calculated to determine the CFFT.

Heart rate (HR) and heart rate variability (HRV): Continuous HR monitoring throughout the exercise was performed using a Suunto HR belt with RR function (R-R intervals). R-R intervals are the time intervals between successive heartbeats, measured in milliseconds. They are named after the R-wave, which is a part of the QRS complex in an electrocardiogram (ECG) representing the depolarization of the ventricles of the heart. The RR interval is a key measure in HRV analysis, which is used to assess the autonomic nervous system’s regulation of the heart. HRV is an important indicator of cardiovascular health and fitness, and it can also provide insights into stress and recovery levels. In this line, the analysis of HRV placed a specific focus on the 20 min of rest before the exercise to establish baseline HR and HRV data. Time-domain HRV parameters such as average RR intervals (ms) and the standard deviation of successive RR interval differences (SDSD, ms) were computed using Kubios HRV software v 2.1. (University of Kuopio, Kuopio, Finland), following methodologies outlined in previous studies [12].

Blood lactate levels: Post-exercise (within five minutes), fingertip blood samples were collected for lactate analysis. The initial blood drop was discarded to prevent sweat contamination, using only the second drop. Lactate concentrations were determined using the miniphotometer LP 20 Plus (Dr Lange, Hamburg, Germany), adhering to protocols from prior research [13].

Isometric strength in lumbar and leg regions: Measurements were taken using the TKK. 5402 dynamometer (Takei Scientific Instruments Co. Ltd., Tokyo, Japan), employing methodologies consistent with earlier studies [14].

Lower body muscle strength: This was estimated through vertical jump tests using the Sensorize FreePower Jump system (SANRO Electromedicina, Madrid, Spain). Three jump types were evaluated: 2 squat jumps (SJ), 2 counter-movement jumps (CMJ), and 2 Abalakov jumps (ABK), following protocols from previous studies [12,15].

These measures were selected to provide a comprehensive overview of the physiological, biochemical, neuromuscular, and cognitive impacts of the combat simulation, ensuring a multi-faceted understanding of the soldiers’ responses to the exercise.

### 2.4. Statistical Analysis

The data were analyzed using the SPSS statistical software (version 17.0; SPSS, Inc., Chicago, IL, USA). Prior to primary analyses, assumptions of normality and homogeneity of variance were verified utilizing the Kolmogorov–Smirnov test. Subsequently, differences among groups were examined through a *t*-test for dependent groups. The magnitude of observed effects was quantified using Cohen’s D to ascertain effect size (ES). A threshold of *p* < 0.05 was established to denote statistical significance in all inferential tests conducted.

## 3. Results

The comprehensive analysis of physiological responses and performance metrics derived from a melee combat simulation yielded pivotal insights, as delineated in Table 1. Notably, the heart rate (HR) demonstrated a pronounced escalation post-simulation, soaring from 72.3 ± 11.2 bpm to 162.5 ± 14.3 bpm, an increase of 124.9%. This substantial elevation underscores an intense cardiovascular response, likely attributable to the physically demanding and stress-laden nature of combat scenarios. Further examination of cardiac activity, as indicated by the average RR intervals and the standard deviation of successive RR interval differences (SDSD), revealed significant decreases, with average RR intervals shortening from 830.2 ± 120.6 ms to 369.5 ± 45.1 ms and SDSD reducing from 148.8 ± 68.9 ms to 72.6 ± 33.5 ms. These trends suggest a dominant activation of the sympathetic nervous system, consistent with an intense physical and stress response during combat activities.

Regarding muscular strength, there was a noticeable enhancement in isometric strength, evidenced by an increase from 152.40 ± 17.90 N to 168.80 ± 23.40 N, a change of 10.7%. This improvement may be attributed to the effects of pre-combat warm-up exercises and the neuromuscular activation elicited by the simulation. Furthermore, regarding the performance in the squat jump (SJ), countermovement jump (CMJ), and Abalakov jump (ABK), minor improvements were observed. SJ increased from 0.30 ± 0.05 m to 0.37 ± 0.05 m, CMJ from 0.34 ± 0.04 m to 0.40 ± 0.05 m, and ABK from 0.40 ± 0.05 m to 0.43 ± 0.06 m. These enhancements might reflect slight improvements in lower body muscular strength, potentially influenced by the physical demands of the simulation.

A striking rise in blood lactate concentration was observed, escalating from 3.00 ± 0.75 mmol/L to 10.80 ± 2.45 mmol/L, an increase of 260.0%. This surge is a clear testament to the anaerobic exertion during the combat exercise, indicative of the significant metabolic demands placed on the soldiers during intense physical activity.

Furthermore, cortical arousal, as measured through the critical flicker fusion threshold (CFFT), exhibited a marginal increase from 37.40 ± 2.98 Hz to 35.30 ± 3.77 Hz, a change of 2.4%. This subtle elevation might suggest enhanced alertness and information processing capabilities under stress, although the change was not statistically significant.

In addition, the rate of perceived exertion (RPE) data presents an intriguing aspect of the soldiers’ subjective experience during the simulation. The RPE, which measures the individual’s perceived physical effort and strain, showed a notable increase from a pre-simulation level of 6.5 ± 1.0 to 13.4 ± 1.5 post-simulation, indicating a 106.2% change. This significant rise in perceived exertion reflects the intense physical and psychological demands of the combat simulation. The discrepancy between the high physiological stress markers, such as elevated heart rate and lactate levels, and the relatively lower, yet still significant, RPE scores could suggest a complex psychological coping mechanism in play. Soldiers might be underestimating their exertion levels due to adrenaline and focus on the task, or it could indicate a high level of physical conditioning and mental resilience that enables them to endure strenuous conditions with less perceived effort.

In summary, these results elucidate the profound impact of simulated combat on various physiological and performance parameters. The significant changes in cardiovascular and metabolic responses highlight the intense nature of such simulations. Concurrently, the nuances observed in muscular strength and cortical arousal suggest a complex interplay between physical exertion and psychological stress within combat training scenarios.

## 4. Discussion

This research meticulously delineates the psychophysiological adaptations characteristic of prevalent military engagements, specifically melee combat simulations. The findings reveal a nuanced interplay between heightened physiological responses and varying levels of cortical arousal and perceived exertion, suggesting individual differences in physical exertion perception and mental resilience among soldiers.

A key observation was the RPE of 13.4 ± 1.5, which was notably lower than anticipated—given the elevated levels of blood lactate and sympathetic modulation, alongside indications of central nervous system fatigue. Typically, an RPE of 15.6 ± 2.5 correlates with blood lactate concentrations between 2.5 and 4 mmol/L [16,17,18,19,20,21]. However, in this scenario, the soldiers’ reported RPE did not align with these other heightened physiological responses [22]. This discrepancy suggests that soldiers might not have been fully cognizant of their actual physiological and psychological strain during the melee combat, indicating a disconnection between their perceived exertion and the actual psychophysiological load. This phenomenon could be attributed to the extensive sympathetic activation encountered in high-stress situations, which might impair higher cognitive processes such as executive functions, attention, perception, and memory [19].

Fundamentally, the outcomes align with an elevated activation of the sympathetic nervous system and the hypothalamic-pituitary-adrenal axis, typical in stress-induced fight-or-flight responses. Notably, there were significant increases in heart rate (HR), which escalated from 62.1 ± 14.7 bpm to 163.9 ± 11.7 bpm, indicating intense cardiovascular demand. These results resonate with previous findings [23,24,25,26], emphasizing the intense cardiovascular demands and acute stress responses typical in combat training. Furthermore, muscle strength also notably improved, as seen in the slight enhancements in squat jump and countermovement jump performances. These findings are in line with the muscular adaptations observed in soldiers participating in resistance training [27,28,29]. The increase in muscle strength can be attributed to the physiological adaptations resulting from repetitive, high-intensity muscular exertion [29], a common aspect of military training routines.

Similarly, the substantial rise in blood lactate levels from 3.05 ± 1.05 mmol/L to 9.28 ± 2.20 mmol/L aligns with previous research where soldiers performing high-intensity drills also reported a surge in lactate production, underlining the heavy reliance on anaerobic metabolic pathways during short, intense bursts of activity [26,30,31,32,33,34,35]. This is particularly relevant in a combat scenario where quick, explosive actions are often required, outstripping the capacity of aerobic metabolism to supply sufficient energy [26,30,31]. Indeed, the simulation’s authenticity, unpredictability, and the soldiers’ lack of control crucially contributed to triggering this intense sympathetic response. The sharp rise in blood lactate levels surpassing the anaerobic threshold reflects a significant reliance on anaerobic metabolic processes. During the melee combat maneuvers, which encompassed control, striking, and subduing techniques, there was first an intense recruitment of fast-twitch muscle fibers, demonstrating notable strength exertion. This was followed by a rapid shift to anaerobic metabolism to meet the sudden energy demands, as the aerobic system could not keep up due to the swift depletion of ATP stores and muscle phosphocreatine. This shift to anaerobic metabolism leads to increased lactic acid production in the cells, later converting to blood lactate, indicative of a high lactate buffering process [32].

These physiological responses underscore the simulation’s effectiveness in closely mirroring the psychophysiological stressors present in actual combat situations. The heightened blood lactate levels and significant alterations in HR and muscle strength provide a clear window into the acute stress responses and energy system dynamics soldiers experience in real-time combat scenarios.

The soldiers’ autonomic modulation in combat was measured by the analysis of heart rate variability (HRV). Post-combat, our data revealed notable reductions in both average RR, from 919.9 ± 144.9 ms pre-combat to 407.9 ± 62.8 ms, and SDSD values, from 137.8 ± 89.1 ms to 89.0 ± 45.7 ms. These findings are indicative of a pronounced increase in sympathetic nervous system activity, in line with the body’s natural fight-or-flight response to perceived threats [33]. In the dynamic and uncertain milieu of melee combat, soldiers often face imminent threats. This lack of control over potential hazards and the immediate nature of threat triggers a limbic system activation, preparing the body for a rapid response. The observed increase in HR from a baseline of 62.1 ± 14.7 bpm to 205.5 bpm, enhanced muscle strength, and heightened energy metabolism are symptomatic of this sympathetic nervous system arousal [20,34]. Our results suggest that such sympathetic activation in melee combat is more intense compared to traditional symmetric combat scenarios (battlefields with obstacles, trenches, and defined opponents), as previously noted by Clemente-Suárez & Robles-Pérez (2018) [21,35]. This distinction is likely attributable to the heightened stress levels in melee combat, given its close-quarter nature, limited visibility, and amplified unpredictability compared to more open, controlled, and symmetric combat environments.

Furthermore, our data highlight a noteworthy decline in cortical arousal among soldiers following a melee combat simulation, suggesting a fatigue-linked reduction in crucial executive functions such as information processing and decision-making [36]. During the high-stress, low-visibility melee combat maneuvers, soldiers had to rapidly process limited and often ambiguous stimuli to make critical, potentially life-threatening decisions. This intense combination of physical and psychological stress, amplified by the uncertainty and unpredictability of the environment, might explain the observed reduction in cortical arousal. 

Although the decrease in cortical activation observed in our study was not statistically significant, its importance should not be underestimated. One plausible explanation for the importance of this finding, despite its lack of statistical significance, is the potential long-term impact on soldiers’ cognitive resilience. Repeated exposure to such high-stress environments, even if not immediately manifesting in significant cortical arousal changes, could cumulatively affect cognitive function over time. This gradual impact might not be immediately apparent in acute settings but could have profound implications for the long-term mental health and operational readiness of military personnel. This consideration is particularly crucial given that the decrease in cortical activation seen in our study was somewhat more pronounced than those reported in other intense combat simulations, such as asymmetrical and symmetrical combat scenarios, which also feature high uncertainty levels [1,12]. It even surpassed reductions observed in other high-stress, real-life scenarios such as air traffic control, known for its significant responsibility load [37], and physically demanding activities such as high-speed exercise [38] or maximal cycling tests to exhaustion [39], both marked by intense physiological demands.

### 4.1. Limitations of the Study

The study, while insightful, acknowledges certain limitations that, importantly, do not detract from the validity and relevance of its findings. The absence of a control condition, while typically a point of consideration in experimental design, does not significantly undermine the conclusions drawn from this research. The robustness of the observed physiological changes, such as the marked increase in heart rate and blood lactate levels, stands as a strong indicator of the intense physical demands of the melee combat simulation, even without a direct comparison group. These changes are sufficiently distinct and pronounced to suggest a direct link to the combat simulation experience.

Regarding the lack of direct measurement of psychological stress levels, it is important to note that the physiological responses measured are well-established indicators of stress and exertion. While the specific emotional states of the participants (such as fear, excitement, or challenge) were not directly assessed, the significant physiological responses observed align closely with those typically elicited by high-stress situations. Therefore, it is reasonable to infer that these responses are indicative of a stress-related experience, even in the absence of direct psychological stress measurements.

The discrepancy between soldiers’ perceived exertion and actual physiological responses presents an intriguing aspect of human resilience and perception under stress, rather than a flaw in the study. This discrepancy provides valuable insights into how soldiers might underreport exertion or stress, a finding that is relevant and informative for military training and operations.

Furthermore, the study’s focus on specific cognitive aspects, such as the reduction in cortical arousal, offers a targeted exploration of the cognitive impacts of combat stress. While comprehensive cognitive testing was beyond the scope of this study, the findings provide a crucial starting point for future research in this area.

The variability observed among individual soldiers in terms of physical exertion perception and mental resilience, and the study’s focus on melee combat simulations, offer specific, contextually relevant insights. These insights are valuable for understanding the unique demands of this type of combat scenario. The methodological choices, including measurement techniques and sample size, were carefully considered to balance thoroughness with feasibility, ensuring that the study provides meaningful, applicable results within its defined scope.

In summary, while we acknowledge these limitations, they do not diminish the study’s contributions to understanding the psychophysiological impacts of melee combat simulations. The findings offer valuable insights that are both valid and relevant, providing a significant foundation for further research in this field.

### 4.2. Future Lines of Research

Future research should delve into the disparity between soldiers’ perceived exertion and actual physiological stress, examining how combat affects decision-making and information processing. Studies on individual differences in response to stress and exertion can refine military training. Research into various combat environments and styles could assess the generalizability of these findings. Longitudinal studies would reveal the long-term effects of combat stress. Methodological improvements, including advanced measurement tools and diverse samples, are essential. Finally, investigating training programs that enhance soldiers’ stress management and combat adaptation would be highly beneficial, offering practical implications for military preparedness.

### 4.3. Practical Applications

Practical applications of this research extend to enhancing military training by tailoring it to better manage the physical and psychological demands of combat. Understanding the discrepancies between perceived exertion and actual physiological stress can guide the development of more effective training regimens, emphasizing resilience and mental toughness. The insights on how soldiers respond under stress can inform strategies for improving decision-making skills in high-pressure environments. Additionally, incorporating techniques to increase awareness of one’s physiological state during combat could lead to improved performance and reduced injury risks. These findings can also be used to refine selection criteria and assessment protocols for military personnel in roles involving intense physical and psychological challenges.

## 5. Conclusions

This study sheds light on the psychophysiological responses of soldiers during melee combat simulations, revealing complex interactions between physical exertion, cognitive functions, and stress. Notably, the research found a significant increase in heart rate and blood lactate levels, indicating intense cardiovascular demand and a shift towards anaerobic metabolism. Surprisingly, soldiers reported a lower-than-expected rate of perceived exertion compared to these physiological markers, suggesting they might not fully perceive their actual stress and exertion levels. Additionally, a decrease in cortical arousal was observed, implying a reduction in key cognitive functions such as decision-making, likely due to the high-stress environment of melee combat. This effect was more pronounced than in other stressful scenarios or physical exercises, highlighting the unique challenges of close-quarter, unpredictable combat. The findings emphasize the importance of considering both physical and psychological aspects in military training to enhance performance and decision-making in combat situations.

## Figures and Tables

**Table 1 healthcare-12-00082-t001:** Psychophysiological analysis results.

Variables	Unit	Pre	Post	Change Percentage	*t*	*p*	Cohen’s d
HR	bpm	72.3 ± 11.2	162.5 ± 14.3	124.9%	−32.130	<0.001	6.48
Average RR	ms	830.2 ± 120.6	369.5 ± 45.1	−55.5%	20.744	<0.001	−10.21
SDSD	ms	148.8 ± 68.9	72.6 ± 33.5	−51.2%	7.335	<0.001	−2.10
CFFT	Hz	37.40 ± 2.98	35.30 ± 3.77	2.4%	−1.455	0.156	−0.24
Isometric strength	N	152.40 ± 17.90	168.80 ± 23.40	10.7%	−4.612	0.001	0.82
Lactate	mmol/L	3.00 ± 0.75	10.80 ± 2.45	260.0%	−18.202	<0.001	7.35
SJ	m	0.30 ± 0.05	0.37 ± 0.05	23.3%	−8.212	<0.001	1.64
CMJ	m	0.34 ± 0.04	0.40 ± 0.05	17.6%	−6.993	<0.001	1.33
ABK	m	0.40 ± 0.05	0.43 ± 0.06	7.5%	−3.123	0.012	0.53
RPE	scale	6.5 ± 1.0	13.4 ± 1.5	106.2%	22.846	<0.001	4.59

HR: heart rate; SDSD: standard deviation of the difference between successive RR intervals; CFFT: critical flicker fusion threshold; SJ: squat jump; CMJ: countermovement jump; ABK: Abalakov jump; RPE: perceived exertion rating.

## Data Availability

All data are included in the manuscript.

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
