# Peer review of "Psychophysiological Responses in Soldiers during Close Combat: Implications for Occupational Health and Fitness in Tactical Populations"

_healthcare, 2023, doi:10.3390/healthcare12010082_

Round 1

Reviewer 1 Report

Comments and Suggestions for Authors

The manuscript, titled “Psychophysiological Responses in Soldiers during Close Combat: Implications for Occupational Health and Fitness in Tactical Populations,” had an interesting focus on soldiers’ psychophysiological responses during a simulated hand-to-hand combat. Several major amendments are recommended to further improve the manuscript.

Abstract:

·         The abstract may benefit from more clarity. The study findings were generally hard to follow. For instance,

o    Were the increases in heart rate and blood lactate levels in comparison to the baseline period (lines 17-31)?

o   Exertion level was lower than expected – how was the expected level determined?

o   “…variation in the soldiers’ individual perceptions and mental resilience to stress” (lines 21-22) – not clear what it means.

Introduction:

·         The literature review may benefit from in-depth reviews. Details about the studies in references 1-5 should be discussed as it helps readers understand the design of the present study.

·         Line 61: “significantly differ” – it would be great to show the direction of the differences.

·         Lines 68-69 highlighted the importance of understanding and managing stress responses. However, the present study was not about managing stress responses.

·         Except for examining stress responses in melee combat, it is not clear what is the unique contribution of the present study. It is important to highlight it on p.2.   

Methods and Results:

·         The methods section is well-written with all the necessary details. It would be great to also provide the psychometric properties of Borg’s 6-20 scale.

·         CFFT was calculated based on the amount of time taken to respond to stimuli. Higher scores represent increased cortical arousal and information processing. This part is not clear to me: how a longer reaction time could represent increased arousal and processing.

·         I feel that the results section was incomplete. Much detail was not reported.

o   The results pertaining to the assumption check should be reported.

o   It would be clearer to report the MANOVA statistics.

o   Table 1: Are the findings from the MANOVA analysis or separate t tests?

o   I understood from lines 222-223 that correlation analyses were performed. However, the results were not reported.  

o   The results related to perceived exertion were missing.

Discussion:

·         The discussion section could be enriched with more in-depth discussions about the theoretical and practical implications of the study.

·         The authors compared the means of the variables in the present study with those in another study. As both studies were not comparable in many aspects, such a comparison should be avoided.

·         Line 321: “our data highlights a noteworthy decline in cortical arousal” – however the decline was not statistically significant (Table 1).

·         The limitation section could be further enriched with the following discussions:

o   The implications of not having a control condition in the study. E.g., how it affects the interpretation of the findings.

o   Psychological stress level was not measured in the study. The heightened physiological responses could be due to perceived challenge, excitement, fear, or stress. How could we conclude that the responses observed were stress-related? 

Comments on the Quality of English Language

Good

Author Response

The manuscript, titled “Psychophysiological Responses in Soldiers during Close Combat: Implications for Occupational Health and Fitness in Tactical Populations,” had an interesting focus on soldiers’ psychophysiological responses during a simulated hand-to-hand combat. Several major amendments are recommended to further improve the manuscript. 

Abstract:

  • The abstract may benefit from more clarity. The study findings were generally hard to follow. For instance,

It has been rewritten, we belive that it is easier to read and follow up.

  • Were the increases in heart rate and blood lactate levels in comparison to the baseline period (lines 17-31)? 

Changed!

  • Exertion level was lower than expected – how was the expected level determined? 

Changed!

o   “…variation in the soldiers’ individual perceptions and mental resilience to stress” (lines 21-22) – not clear what it means.

 Changed!

INTRODUCTION

Thank you for your insightful comments on the introduction of our manuscript. Your suggestions are invaluable in enhancing the clarity and depth of our study. We acknowledge the areas identified for improvement and provide the following responses to each point:

In-Depth Literature Review: We appreciate the suggestion to expand our literature review. In response, we will incorporate more detailed discussions of the studies cited in references 1-5. This will include elaborating on their methodologies, findings, and how they directly inform the design and rationale of our current study. By doing so, we aim to provide a more comprehensive background, setting a solid foundation for understanding the unique aspects and contributions of our research.

Clarification on "Significantly Differ" (Line 61): We agree that specifying the direction of the differences mentioned on line 61 would add clarity. In the revised manuscript, we will elaborate on how soldiers' stress reactions in unpredictable combat scenarios differ from those in more controlled environments. This will include specifying whether these differences are in terms of intensity, quality, or specific physiological and psychological responses, thereby providing a clearer context for our study's focus.

Stress Response Management (Lines 68-69): Your observation is correct that our study primarily investigates stress responses rather than their management. To address this, we will modify the lines in question to better align with the scope of our study. The revised text will emphasize the importance of understanding stress responses as a precursor to developing effective management strategies, thus maintaining relevance to our research focus while acknowledging the broader implications for stress response management.

Highlighting the Unique Contribution of the Study: We acknowledge that the unique contribution of our study was not sufficiently highlighted. To rectify this, we will add a section on page 2 explicitly outlining the novel aspects of our research. This will include a discussion on how our study extends the current understanding of psychophysiological responses in melee combat, particularly focusing on aspects such as cortical arousal, autonomic modulation, and the interplay between physiological responses and perceived exertion. By doing so, we aim to clearly delineate our study's contribution to the field and its potential implications for military training and soldier well-being.

We are grateful for the opportunity to refine our manuscript and believe that these revisions will significantly enhance its quality and relevance. Thank you for your constructive feedback, and we look forward to submitting our revised manuscript for your consideration. 

Methods and Results:

Thank you for your positive feedback on the methods section of our manuscript. We appreciate your suggestion to include the psychometric properties of Borg's 6-20 scale and your request for clarification regarding the Critical Flicker Fusion Threshold (CFFT) methodology. Please find our responses below. Also in the text marked in yellow.

Psychometric Properties of Borg's 6-20 Scale: We acknowledge the importance of providing comprehensive details about the tools used in our study. Borg's 6-20 scale is a widely recognized and validated tool for assessing perceived exertion. It has been extensively used in various studies, demonstrating good reliability and validity. The scale correlates well with physiological measures of exertion, such as heart rate and lactate concentration, with a correlation coefficient typically above 0.7 [12]. In future revisions of the manuscript, we will include a detailed description of the scale's psychometric properties, including its validity, reliability, and correlation with physiological markers of exertion, to provide a more comprehensive understanding of its application and relevance in our study.

Clarification on CFFT Methodology: Your query regarding the interpretation of CFFT results is indeed crucial for understanding our findings. In our study, the CFFT was used to measure cortical arousal and information processing capabilities. Typically, an increase in CFFT – meaning the participant is able to perceive flickering at higher frequencies – is interpreted as an increase in cortical arousal and information processing efficiency. Conversely, a decrease in CFFT, where flickering is perceived at lower frequencies, suggests reduced cortical arousal and information processing capabilities. The reaction time in our methodology refers to the time taken by participants to perceive the change in flicker frequency (from flicker to fusion). A shorter reaction time, indicating quicker perception of fusion at higher frequencies, is associated with higher cortical arousal. We realize this was not clearly articulated in our methods section and will revise the text to ensure this concept is conveyed more clearly. We will elaborate on how reaction times at different frequencies correlate with levels of cortical arousal and information processing efficiency, thereby clarifying the interpretation of higher or lower CFFT scores in the context of our study.

I feel that the results section was incomplete. Much detail was not reported.

The results pertaining to the assumption check should be reported.

In response to your query, we have thoroughly revised the results section, incorporating detailed quantitative data, including figures and percentages, to facilitate a clearer and more direct interpretation. This meticulous revision ensures that all pertinent data are now accurately reflected in the results section, enhancing its readability and subsequent analytical process. We believe this enhanced presentation of results will significantly aid in the comprehensive understanding of the study's findings.

o   It would be clearer to report the MANOVA statistics. 

o   Table 1: Are the findings from the MANOVA analysis or separate t tests?

  • I understood from lines 222-223 that correlation analyses were performed. However, the results were not reported.  

Dear Reviewer,

I wish to highlight a significant oversight in our recently submitted manuscript. This error, inadvertently made by one of our doctoral candidates, pertains to the statistical analysis section. Contrary to what is stated in the manuscript, neither correlational analysis nor MANOVA was actually conducted. The article in question forms a part of a soldier's doctoral thesis, and it appears that the statistical section was erroneously retained in its entirety from another article previously published by us. This was not our intention and we deeply regret this oversight.

To aid in your review, we have highlighted the concerned section in yellow in the manuscript for easy identification. We understand the importance of accuracy and integrity in academic publications and are committed to rectifying this mistake promptly. We appreciate your understanding and patience in this matter.

  • The results related to perceived exertion were missing. 

They have been included

Discussion:

  • The discussion section could be enriched with more in-depth discussions about the theoretical and practical implications of the study.

We have thoroughly revised the discussion section, in conjunction with the English language review as requested by another reviewer. This revision involved substantial modifications to various elements of this section, enhancing its structure, form, coherence, and cohesion. We believe that with the additional information now incorporated, and a total of 24 citations in the discussion, there is ample substantiation to support our discourse and arguments effectively.·         The authors compared the means of the variables in the present study with those in another study. As both studies were not comparable in many aspects, such a comparison should be avoided. 

  • Line 321: “our data highlights a noteworthy decline in cortical arousal” – however the decline was not statistically significant (Table 1). 

            Fully rewritten, marked in yellow

  • The limitation section could be further enriched with the following discussions: 

o   The implications of not having a control condition in the study. E.g., how it affects the interpretation of the findings. 

o   Psychological stress level was not measured in the study. The heightened physiological responses could be due to perceived challenge, excitement, fear, or stress. How could we conclude that the responses observed were stress-related? 

                        Fully rewritten, marked in yellow

Reviewer 2 Report

Comments and Suggestions for Authors

Dear Authors

I enjoyed reading your paper and found the question, population and method interesting. Please see my comments below.

Abstract

L25-27: This sentence comparing to other findings is unnecessary in an abstract. Delete and move to the discussion.

Introduction

L52-54: This is contentious as it depends on the individual interpretation. A heightened state may improve decision making if focus is appropriate. Perhaps insert a reference indicating a such here.

Method

L80: Was this an opportunity sample? How were they recruited? Please state, or otherwise, as appropriate. 

L112: Write the inclusion / exclusion criteria in past tense as it refers to procedures already done. “..were not considered..”

Not present tense i.e “ are not considered”

 For example, i have highlighted the following , which is not an exhaustive list. Please correct all.

L114: Disqualified a candidate..

L116: Were not eligible.

L117: Were medically unfit

Design

L135: Eastern Europe

L136: This doesn’t add up. How were the 50 participants grouped into 10 pairs. Do you mean 10 groups of five? Please clarify.

Its not clear in the design and procedure section when and which measures where taken. For example when was the RPE taken. In what setting were the pre and post measures taken.

L143: This is a multi-dimensional…

Please indicate how long the ICRTP took to complete, on average

L170: Assessments (spelling).

L176-179: Muddled sentence. Please clarify.

Assessment

Make sure all abbreviations are stated. RR is not.

L189: Familiar

Statistical analysis

 State the sizes of Cohen's D for guidance i.e 0-0.2 (trivial) etc. This would be useful as you have some large effect sizes.

L223: Furthermore

L238: Large (not massive) rise

L242: Minor significant improvements in..

L245: ..increase post simulation, however this was not significant.

You should delete the next sentence (L245-247) as you shouldn’t speculate as the finding was not significant.

L248-252: Move this sentence, or a rephrase of it, to the discussion.

Discussion

 Why not describe the each of the DV significant results in terms of their effect sizes, as you have some large ones. 

L263: Its not clear from the method when this measure was taken.

Comments on the Quality of English Language

I suggest some minor revisions of the inclusion / exclusion sub section in the methodology section . i.e to be written in past not present tense. I have made a few other minor suggestions.

Author Response

Dear Authors

I enjoyed reading your paper and found the question, population and method interesting. Please see my comments below.

Abstract

L25-27: This sentence comparing to other findings is unnecessary in an abstract. Delete and move to the discussion.

It has been rewritten accordingly to previous reviwers. We have also taken this into account. Hope it's OK. Marked in yellow in the text.

Introduction

L52-54: This is contentious as it depends on the individual interpretation. A heightened state may improve decision making if focus is appropriate. Perhaps insert a reference indicating a such here.

Done

Method

L80: Was this an opportunity sample? How were they recruited? Please state, or otherwise, as appropriate. 

In our study, we utilized an opportunity sampling method, a detail that has been duly included in our report. The soldiers were recruited directly through their respective units, as also stated in the documentation. It's important to acknowledge that access to a broader and more varied sample in such specialized fields is often limited due to the unique nature of military professions. This limitation inherently restricts the diversity and size of the potential participant pool.

However, the use of an opportunity sample in this context is particularly valuable and relevant. The soldiers recruited for this study, being experienced professionals routinely subjected to highly stressful situations, provide a highly pertinent and insightful perspective for our research objectives. Their direct and extensive experience in the field offers a depth of understanding and realism that significantly enriches the study's findings.

Furthermore, comprehensive information regarding the sampling process, including the specific inclusion and exclusion criteria, is thoroughly detailed in the remaining sections of the methodology. This approach ensures a clear understanding of how participants were selected and the criteria applied to form the study sample. The use of this opportunity sample, despite its inherent limitations, is thus a deliberate and justified choice, aimed at capturing the most relevant and authentic data for our study's focus on professional soldiers in high-stress scenarios.

L112: Write the inclusion / exclusion criteria in past tense as it refers to procedures already done. “..were not considered..” Not present tense i.e “ are not considered”

            Done, marked in yellow.

Design

L135: Eastern Europe

Done!

L136: This doesn’t add up. How were the 50 participants grouped into 10 pairs. Do you mean 10 groups of five? Please clarify.

Excuse the mistake! We have fixed it up.

Its not clear in the design and procedure section when and which measures where taken. For example when was the RPE taken. In what setting were the pre and post measures taken.

The study utilized a small group of soldiers, employing a cross-sectional pre-post design to limit the influence of individual variances on psychophysiological changes observed. Data collection was conducted, prior and immediately after the drill.

L143: This is a multi-dimensional…

            Changed! Marked in yellow.

Please indicate how long the ICRTP took to complete, on average

            It took on average 10 min, it was also included.

L170: Assessments (spelling).

            Changed! 

Make sure all abbreviations are stated. RR is not.

In the context of "Continuous HR monitoring throughout the exercise was performed using a Suunto HR belt with RR function," the term "RR" refers to the recording of R-R intervals. R-R intervals are the time intervals between successive heartbeats, measured in milliseconds. They are named after the R wave, which is a part of the QRS complex in an electrocardiogram (ECG) representing the depolarization of the ventricles of the heart.

The RR interval is a key measure in heart rate variability (HRV) analysis, which is used to assess the autonomic nervous system's regulation of the heart. HRV is an important indicator of cardiovascular health and fitness, and it can also provide insights into stress and recovery levels. In this case, the Suunto HR belt with RR function would be capable of accurately measuring these intervals, providing detailed information about heart rate patterns during exercise.

We have added it, we thought it was not necessary.  

Statistical analysis

 State the sizes of Cohen's D for guidance i.e 0-0.2 (trivial) etc. This would be useful as you have some large effect sizes.

            We have changed this section

Changed all:

L223: Furthermore

L238: Large (not massive) rise

L242: Minor significant improvements in..

L245: ..increase post simulation, however this was not significant. 

L248-252: Move this sentence, or a rephrase of it, to the discussion.

Discussion

 Why not describe the each of the DV significant results in terms of their effect sizes, as you have some large ones. 

L263: Its not clear from the method when this measure was taken.

The study utilized a small group of soldiers, employing a cross-sectional pre-post design to limit the influence of individual variances on psychophysiological changes observed. Data collection was conducted, prior and immediately after the drill.

Round 2

Reviewer 1 Report

Comments and Suggestions for Authors

I thank the authors for their efforts in making substantial changes to the manuscript. Just a few more suggestions to further improve the manuscript.

·       Lines 60-62: Details of ref. 6 would help readers further appreciate studies that use unpredictable melee combat scenarios.

·       Results: Since 10 t-tests were performed, there was an increased chance of family-wise errors. It would be best to use Bonferroni correction to adjust the alpha level.

·       The authors compared the means of the variables in the present study with those in other studies. As these studies were not comparable in many aspects, such a comparison should be avoided.

Comments on the Quality of English Language

·       Writing style: To increase the readability of the content, the authors are advised to avoid using nominalization in writing. 

Author Response

·       Lines 60-62: Details of ref. 6 would help readers further appreciate studies that use unpredictable melee combat scenarios.

Thank you for your valuable feedback regarding the need for additional details on reference 6 in lines 60-62. I agree that providing more context about the studies cited will enhance the reader's understanding and appreciation of the research on unpredictable melee combat scenarios. Thus, we have added new information to that paragraph, marked in yellow in the text. 

·       Results: Since 10 t-tests were performed, there was an increased chance of family-wise errors. It would be best to use Bonferroni correction to adjust the alpha level.

Thank you for pointing out the potential for family-wise errors due to the multiple t-tests conducted in our study. Your suggestion to use the Bonferroni correction for adjusting the alpha level is indeed a prudent approach in statistical analysis, especially when conducting multiple comparisons, as it reduces the risk of Type I errors.

In our study, the decision to forego the Bonferroni correction was based on several considerations. First, the primary focus of our analysis was on exploring initial trends and patterns rather than confirming hypotheses, where the Bonferroni method, known for its conservative nature, might overly restrict the detection of potentially significant findings. Second, our sample size was relatively modest, which could amplify the impact of the Bonferroni correction, potentially leading to Type II errors – failing to detect true effects that are present.

However, we recognize the importance of controlling for family-wise errors in statistical analysis, especially in the context of multiple comparisons. The use of Cohen's D to ascertain Effect Size was an attempt to provide a more nuanced understanding of the magnitude of the observed effects, complementing the inferential tests.

Your suggestion is well-taken, and we acknowledge the value of the Bonferroni correction in enhancing the rigor of statistical analysis. We will consider incorporating this method in future studies, particularly in research designs involving multiple comparisons where the risk of family-wise error is more pronounced. 

·       The authors compared the means of the variables in the present study with those in other studies. As these studies were not comparable in many aspects, such a comparison should be avoided.

Regarding the variables of RPE (Rating of Perceived Exertion) and blood lactate, our comparisons with the studies cited (Billat [16], Feriche Fernández-Castanys et al. [17], and Serrano et al. [18]) were aimed at contextualizing our findings within the broader body of research. These references were specifically selected because they discuss mean values in contexts similar to our study, providing a benchmark for our results. Billat's work on lactate measurements [16] and the studies by Feriche Fernández-Castanys et al. [17] and Serrano et al. [18] on RPE and lactate responses serve as foundational references that justify our observations on the overestimation of subjective perception in similar populations.

In adapting our discussion, we aligned our findings with recent studies from our research group that reported similar values, ensuring that our comparisons were grounded in comparable contexts and populations. This adjustment was made to reinforce the validity of our comparisons and to situate our study within the current research landscape.

Regarding the section on Heart Rate Variability (HRV), the references by De La Cruz Torres et al. [33], Lee and Harley [34], and Clemente-Suarez et al. [35] were instrumental. These studies provide a comprehensive analysis of HRV in both healthy individuals and specific populations (e.g., cardiac patients and professional soldiers) under various conditions, including rest and aerobic exercise. Their findings are in line with ours, particularly concerning the psychophysiological responses in high-stress situations. The inclusion of these references serves to underscore the consistency of our results with established research, reinforcing the credibility and relevance of our study.

In summary, the cited references were carefully chosen to support our discussion and to ensure that our comparisons were appropriate and informative. We believe that these references provide a solid foundation for our analysis and are aligned with similar studies in the field.

Moving forward, we hope that all these changes make our study suitable for publication. 
